# Selecting likely causal risk factors from high-throughput experiments using multivariable Mendelian randomization

Verena Zuber [1,2], Johanna Maria Colijn [3,4], Caroline Klaver [3,4,5] & Stephen Burgess [1,6]*

Modern high-throughput experiments provide a rich resource to investigate causal determinants of disease risk. Mendelian randomization (MR) is the use of genetic variants as instrumental variables to infer the causal effect of a specific risk factor on an outcome. Multivariable MR is an extension of the standard MR framework to consider multiple potential risk factors in a single model. However, current implementations of multivariable MR use standard linear regression and hence perform poorly with many risk factors. Here, we propose a two-sample multivariable MR approach based on Bayesian model averaging (MR-BMA) that scales to high-throughput experiments. In a realistic simulation study, we show that MR-BMA can detect true causal risk factors even when the candidate risk factors are highly correlated. We illustrate MR-BMA by analysing publicly-available summarized data on metabolites to prioritise likely causal biomarkers for age-related macular degeneration.

[1] MRC Biostatistics Unit, School of Clinical Medicine, University of Cambridge, Cambridge, UK. [2] Department of Epidemiology and Biostatistics, Imperial College London, London, UK. [3] Department of Epidemiology, Erasmus University Medical Center, Rotterdam, The Netherlands. [4] Department of Ophthalmology, Erasmus University Medical Center, Rotterdam, The Netherlands. [5] Department of Ophthalmology, Radboud University Medical Center, Nijmegen, The Netherlands. [6] MRC/BHF Cardiovascular Epidemiology Unit, School of Clinical Medicine, University of Cambridge, Cambridge, UK. *email: sb452@medschl.cam.ac.uk

Mendelian randomization (MR) is the use of genetic variants to infer the presence or absence of a causal effect of a risk factor on an outcome. Under the assumption that the genetic variants are valid instrumental variables, this causal effect can be consistently inferred even in the presence of unobserved confounding factors[1]. The instrumental variable assumptions are illustrated by a directed acyclic graph as shown in Fig. 1[2].

Recent years have seen an explosion in the size and scale of data sets with biomarker data from high-throughput experiments and concomitant genetic data. These biomarkers include proteins[3], blood cell traits[4], metabolites[5] or imaging phenotypes such as from cardiac image analysis[6]. High-throughput experiments provide ideal data resources for conducting MR investigations in conjunction with case-control data sets providing genetic associations with disease outcomes (such as from CARDIo-GRAMplusC4D for coronary artery disease[7], DIAGRAM for type 2 diabetes[8], or the International Age-related Macular Degeneration Genomics Consortium [IAMDGC] for age-related macular degeneration[9]). In addition to their untargeted scope, one specific feature of high-throughput experiments is a distinctive correlation pattern between the candidate risk factors shaped by latent biological processes.

Multivariable MR is an extension of standard (univariable) MR that allows multiple risk factors to be modelled at once[10]. Whereas univariable MR makes the assumption that genetic variants specifically influence a single risk factor, multivariable MR makes the assumption that genetic variants influence a set of multiple measured risk factors and thus accounts for measured pleiotropy. Our aim is to use genetic variation in a multivariable MR paradigm to select which risk factors from a set of related and potentially highly correlated candidate risk factors are causal determinants of an outcome. Existing methods for multivariable MR are designed for a small number of risk factors and do not scale to the dimension of high-throughput experiments. We therefore seek to develop a method for multivariable MR that can select and prioritize biomarkers from high-throughput experiments as risk factors for the outcome of interest. In this context we propose a Bayesian model averaging approach (MR-BMA) that scales to the dimension of high-throughput experiments and enables risk factor selection from a large number of candidate risk factors. MR-BMA is formulated on two-sample summarized genetic data which is publicly available and allows the sample size to be maximized.

To illustrate our approach, we analyse publicly available summarized data from a metabolite genome-wide association study (GWAS) on nearly 25,000 participants to rank and prioritise metabolites as potential biomarkers for age-related macular degeneration. Data are available on genetic associations with 118 circulating metabolites measured by nuclear magnetic resonance (NMR) spectroscopy[11] from http://computationalmedicine.fi/data#NMR_GWAS. This NMR platform provides a detailed characterisation of lipid subfractions, including 14 size categories of lipoprotein particles ranging from extra small (XS) high density lipoprotein (HDL) to extra-extra-large (XXL) very low density lipoprotein (VLDL). For each lipoprotein category, measures are available of total cholesterol, triglycerides, phospholipids, and cholesterol esters, and additionally the average diameter of the lipoprotein particles. Apart from lipoprotein measurements, this metabolite GWAS estimated genetic associations with amino acids, apolipoproteins, fatty and fluid acids, ketone bodies and glycerides. We assess the performance of our proposed method in a simulation study with scenarios motivated by the metabolite GWAS and by publicly available summary data on blood cell traits measured on nearly 175,000 participants[4].

## Results

**Multivariable Mendelian randomization and risk factor selection.** Standard MR requires genetic variants to be specific in their associations with a single risk factor of interest, and does not allow genetic variants to have pleiotropic effects on other risk factors on competing causal pathways. Multivariable MR allows genetic variants to be associated with multiple risk factors, provided these risk factors are measured and included in the analysis. Hence multivariable MR allows for 'measured pleiotropic effects'[10,12]. As illustration, multivariable MR can be considered as an extension of the standard MR paradigm (Fig. 1) to model not one, but multiple risk factors (Fig. 2).

We consider a two-sample framework, where the genetic associations with the outcome (sample 1) are regressed on the genetic associations with all the risk factors (sample 2) in a multivariable regression which is implemented in an inverse-variance weighted (IVW) linear regression. Each genetic variant contributes one data point (or observation) to the regression model. Weights in this regression model are proportional to the inverse of the variance of the genetic association with the outcome. This is to ensure that genetic variants having more precise association estimates receive more weight in the analysis. The causal effect estimates from multivariable MR represent the direct causal effects of the risk factors in turn on the outcome when all the other risk factors in the model are held constant[12,13] and Supplementary Fig. 1). Including multiple risk factors into a single model allows genetic variants to have pleiotropic effects on the risk factors in the model referred to as "measured pleiotropy"[14].

However, the current implementation of multivariable MR is not designed to consider a high-dimensional set of risk factors and is not suitable to select biomarkers from high-throughput experiments.

To allow joint analysis of biomarkers from high-throughput experiments in multivariable MR, we cast risk factor selection as

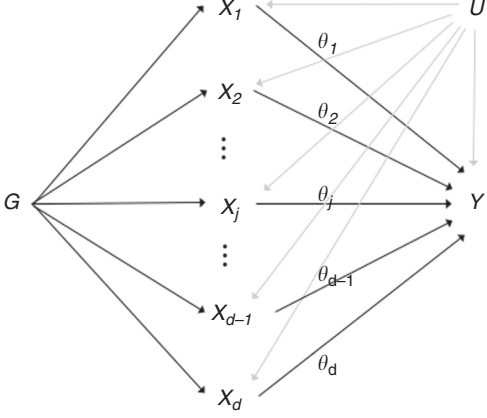

**Fig. 2 Directed acyclic graph of instrumental variable assumptions made in multivariable Mendelian randomization.** $G$ = genetic variants, $X_j$ = risk factor $j$ for $j = 1, \ldots, d$, $Y$ = outcome, $U$ = confounders, $\theta_j$ = causal effect of risk factor $j$.

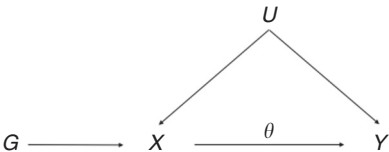

**Fig. 1 Directed acyclic graph of instrumental variable assumptions made in univariable Mendelian randomization.** $G$ = genetic variant(s), $X$ = risk factor, $Y$ = outcome, $U$ = confounders, $\theta$ = causal effect of interest.

variable selection in the same weighted linear regression model as in the IVW method. Formulated in a Bayesian framework (for full details we refer to the Methods section) we use independence priors and closed-form Bayes factors to evaluate the posterior probability (PP) of specific models (i.e. one risk factor or a combination of multiple risk factors). In high-dimensional variable selection, the evidence for one particular model can be small because the model space is very large and many models might have comparable evidence. This is why MR-BMA uses Bayesian model averaging (BMA) and computes for each risk factor its marginal inclusion probability (MIP), which is defined as the sum of the posterior probabilities over all models where the risk factor is present. MR-BMA reports the model-averaged causal effects (MACE), representing conservative estimates of the direct causal effect of a risk factor on the outcome averaged across these models. These estimates can be used to compare risk factors or to interpret effect directions, but should not be interpreted absolutely. As we show in a simulation study based on real biomarker data, MR-BMA provides effect estimates biased towards the Null when there is a causal effect but reduces the variance of the estimate, trading bias for reduced variance. Consequently, MR-BMA enables a better and more stable detection of the true causal risk factors than either the conventional IVW method or other variable selection methods.

**Detection of invalid and influential instruments**. Invalid instruments may be detected as outliers with respect to the fit of the linear model. Outliers may arise for a number of reasons, but they are likely to arise if a genetic variant has an effect on the outcome that is not mediated by one or other of the risk factors—an unmeasured pleiotropic effect. To quantify outliers we use the $Q$-statistic, which is an established tool for identifying heterogeneity in meta-analysis[15]. More precisely, to pinpoint specific genetic variants as outliers we use the contribution $q$ of the variant to the overall $Q$-statistic, where $q$ is defined as the weighted squared difference between the observed and predicted association with the outcome.

Even if there are no outliers, it is advisable to check for influential observations and re-run the approach omitting that influential variant from the analysis. If a particular genetic variant has a strong association with the outcome, then it may have undue influence on the variable selection, leading to a model that fits that particular observation well, but other observations poorly. To quantify influential observations, we suggest to use Cook's distance ($Cd$)[16]. We illustrate the detection of influential points and outliers in the applied example and provide more details in the Methods.

**Simulation results**. To assess the performance of the proposed method, we perform a simulation study in three scenarios based on real high-dimensional data. We compare the performance of the conventional approach (Multivariable IVW regression), the Lars[17], Lasso, and Elastic Net[18] penalised regression methods developed for high-dimensional regression models, MR-BMA, and the model with the highest posterior probability from the BMA procedure (best model). We seek to evaluate two aspects of the methods: (1) how well can the methods select the true causal risk factors, and (2) how well can the methods estimate causal effects. Risk factor selection is evaluated using the receiver operating characteristic (ROC) curve, where the true positive rate is plotted against the false positive rate. True positives are defined as the risk factors in the generation model that have a non-zero causal effect. Causal estimation is evaluated by calculating the mean squared error (MSE) of estimates, which is defined as the squared difference between the estimated causal effect and the

true causal effect. The MSE of an estimator decomposes into the sum of its squared bias and its variance.

Genetic associations with the risk factors are obtained from three different scenarios. Two scenarios are based on the NMR metabolite GWAS by ref. [11], where we use as instrumental variables $n = 150$ independent genetic variants that were associated with any of three composite lipid measurements (LDL cholesterol, triglycerides or HDL cholesterol) at a genome-wide level of significance ($p < 5 \times 10^{-8}$) in a large meta-analysis of the Global Lipids Genetics Consortium[19]. In Scenario 1, we consider a small set of $d = 12$ randomly selected risk factors, and in Scenario 2 a larger set of $d = 92$ risk factors. Scenario 3 is based on publicly available summary data on $d = 33$ blood cell traits measured on nearly 175,000 participants[4]. Using all genetic variants that were genome-wide significant for any blood cell trait, we have $n = 2667$ genetic variants as instrumental variables. For each scenario, we generate the genetic associations for the outcome based on four random risk factors having a positive effect in Setting A and on eight random risk factors, of which four have a positive and four have a negative effect, in Setting B. In addition, we vary the proportion of variance in the outcome explained by the causal risk factors. Each simulation setting is repeated 1000 times. Full detail of the generation of the simulated outcomes is given in the Supplementary Methods.

Looking at a small set of $d = 12$ risk factors in the NMR metabolite data of which four risk factors are true causal ones (Scenario 1, Setting A), we see that MR-BMA is dominating all other methods in terms of area under the ROC curve (see Fig. 3a). Next best methods are Lasso, Elastic Net, the Bayesian best model and Lars. The standard IVW method gives the worst performance. Similar results were obtained when varying the variance in the outcome explained by the risk factors (Setting A in Supplementary Fig. 3 and Setting B in Supplementary Fig. 4). With respect to the MSE of estimates (Table 1), MR-BMA has the lowest MSE in almost all scenarios followed by Elastic Net, Lasso, the Bayesian best model, and then Lars. Elastic Net has the lowest MSE for $R^2 = 0.5$ in setting B. The highest MSE is seen for the IVW method, which provides unbiased estimates, as can be seen in Supplementary Figs. 5 and 6, but at the price of a high variance. As can be seen from Supplementary Table 1, all estimation methods except the IVW are biased conservatively towards the Null when there is a true causal effect. Yet, all causal effect estimates are unbiased when there is no causal effect. Supplementary Table 2 (Setting A) and Supplementary Table 3 (Setting B) provide the mean and the standard deviation of the causal effect estimates, which confirm the large standard deviation of the IVW estimate compared with the other approaches.

When increasing the number of risk factors to $d = 92$ while keeping the number of true causal risk factors constant to four (Scenario 2, Setting A), the standard IVW method fails to distinguish between true causal and false causal risk factors and provides a ranking of risk factors which is nearly random as shown in the ROC curve in Fig. 3b and Supplementary Figs. 7 and 8. Despite being unbiased (see Supplementary Figs. 9 and 10, Supplementary Tables 1, 2 and 3), the variance of the IVW estimates is large and prohibits better performance. In contrast, Lars, Lasso, Elastic Net and MR-BMA provide causal estimates which are biased towards zero, but have much reduced variance compared with the IVW estimates. The Lasso provides sparse solutions with many of the causal estimates set to zero. This allows the Lasso and Elastic Net to have relatively good performance at the beginning of the ROC curve, but their performance weakens when considering more risk factors. The best performance in terms of the ROC characteristics is observed for MR-BMA. In terms of MSE (Table 1), the dominant role of the variance of the IVW estimate becomes

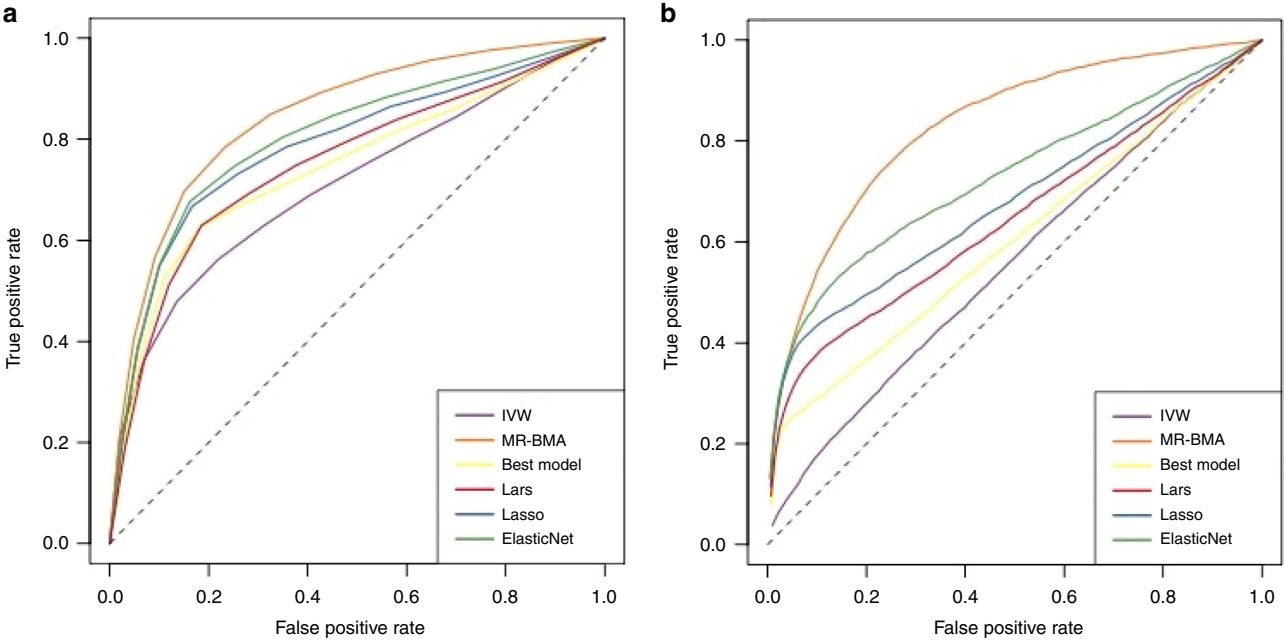

**Fig. 3 Receiver operating characteristic (ROC) curve for simulation study on metabolite GWAS.** ROC curves plotting the true positive rate against the false positive rate for **a** a small number of risk factors ($d = 12$) of which four have true positive effects (Scenario 1, Setting A) and for **b** a large number of risk factors ($d = 92$) of which four have true positive effects (Scenario 2, Setting A). Proportion of variance explained ($R^2$) is set to 0.3.

**Table 1 Mean squared error (MSE) of the causal effect estimates from the competing methods on the NMR metabolite and blood cell trait data.**

| | Setting A | | | Setting B | | |
|---|---|---|---|---|---|---|
| $R^2$: | 0.1 | 0.3 | 0.5 | 0.1 | 0.3 | 0.5 |
| Scenario 1: | | | | | | |
| IVW | 0.6727 | 0.1675 | 0.0784 | 0.5949 | 0.1619 | 0.0629 |
| Lars | 0.1292 | 0.0447 | 0.0298 | 0.1559 | 0.0648 | 0.0372 |
| Lasso | 0.0604 | 0.0289 | 0.0162 | 0.1046 | 0.0503 | 0.0307 |
| Elastic Net | 0.0673 | 0.0300 | 0.0162 | 0.1161 | 0.0480 | **0.0287** |
| MR-BMA | **0.0340** | **0.0175** | **0.0105** | **0.0534** | **0.0368** | 0.0306 |
| Best model | 0.0717 | 0.0320 | 0.0156 | 0.0921 | 0.0514 | 0.0376 |
| Scenario 2: | | | | | | |
| IVW | 22.9516 | 6.0594 | 2.6257 | 23.2495 | 5.7715 | 2.4802 |
| Lars | 0.0354 | 0.0367 | 0.0094 | 0.0321 | 0.0212 | 0.0143 |
| Lasso | 0.0064 | 0.0047 | 0.0039 | 0.0105 | 0.0086 | 0.0074 |
| Elastic Net | 0.0064 | 0.0044 | 0.0034 | 0.0098 | 0.0078 | 0.0067 |
| MR-BMA | **0.0051** | **0.0039** | **0.0032** | **0.0088** | **0.0076** | **0.0063** |
| Best model | 0.0114 | 0.0081 | 0.0061 | 0.0150 | 0.0121 | 0.0096 |
| Scenario 3: | | | | | | |
| IVW | 1.6200 | 0.4272 | 0.1742 | 2.3140 | 0.6208 | 0.2566 |
| Lars | 0.3461 | 0.1151 | 0.0482 | 0.5892 | 0.1669 | 0.0844 |
| Lasso | 0.0161 | 0.0067 | 0.0040 | 0.0378 | 0.0225 | 0.0166 |
| Elastic Net | 0.0168 | 0.0074 | 0.0044 | 0.0451 | 0.0224 | 0.0169 |
| BMA | **0.0066** | **0.0034** | **0.0019** | **0.0235** | **0.0165** | **0.0149** |
| Best model | 0.0128 | 0.0051 | 0.0027 | 0.0444 | 0.0242 | 0.0177 |

We mark in bold font the lowest MSE in each experimental setting. Scenario 1: NMR metabolites, $d = 12$ risk factors, Scenario 2: NMR metabolites, $d = 92$ risk factors, and Scenario 3: blood cell traits, $d = 33$ risk factors. Setting A includes four true causal risk factors which increase the risk and Setting B includes eight true causal risk factors of which half are protective and the other half increases the risk

again apparent as the IVW method has a thousand times larger MSE than MR-BMA, which has the lowest MSE for all scenarios considered. Similarly to earlier results on the bias of the effect estimates we find that the IVW is unbiased when there is a causal effect, while the other methods designed for high-dimensional settings are conservatively biased towards the null, and only unbiased when there is no causal effect (Supplementary Table 1).

In the blood cell trait data (Scenario 3), MR-BMA has again the lowest MSE, followed by the regularised regression approaches

and the best model in the Bayesian approach. Despite a large sample size ($n = 2667$) and comparatively low dimension of the risk factor space ($d = 33$), the IVW approach is the only unbiased method at the cost of an inferior detection of true positive risk factors (Supplementary Figs. 12 and 13) and a large variance (Supplementary Figs. 14 and 15, Supplementary Tables 2 and 3), and consequently a MSE which is in a magnitude of a hundred larger than other methods designed for high-dimensional data analysis (Table 1).

**Metabolites as risk factors for age-related macular degeneration**. Next we demonstrate how MR-BMA can be used to select metabolites as causal risk factors for age-related macular degeneration (AMD). AMD is a painless eye-disease that ultimately leads to the loss of vision. AMD is highly heritable with an estimated heritability of up to 0.71 for advanced AMD in a twin study[20]. A GWAS meta-analysis has identified 52 independent common and rare variants associated with AMD risk at a level of genome-wide significance[9]. Several of these regions are linked to lipids or lipid-related biology, such as the *CETP*, *LIPC* and *APOE* gene regions[21]. Lipid particles are deposited within drusen in the different layers of Bruch's membrane in AMD patients[21]. A recent observational study has highlighted strong associations between lipid metabolites and AMD risk[22].

This evidence for lipids as potential risk factor for AMD has motivated a multivariable MR analysis which has shown that HDL cholesterol may be a putative risk factor for AMD, while there was no evidence of a causal effect for LDL cholesterol and triglycerides[23]. Here, we extend this analysis to consider not just three lipid measurements, but a wider and more detailed range of $d = 30$ metabolite measurements to pinpoint potential causal effects more specifically. As summary-level data we use $d = 30$ metabolites as measured in the metabolite GWAS described earlier[11] for the same lipid-related instrumental variants as described previously. All of these metabolites have at least one genetic variant used as an instrumental variable that is genome-wide significant and no genetic associations of metabolites are stronger correlated than $r = 0.985$. First, we prioritise and rank risk factors by their marginal inclusion probability (MIP) from MR-BMA using $\sigma^2 = 0.25$ as prior variance and $p = 0.1$ as prior probability, corresponding to a priori three expected causal risk factors. Secondly, we perform model diagnostics based on the best models with posterior probability >0.02.

When including all genetic variants available in both the NMR and the AMD summary data ($n = 148$), the top risk factor with respect to its MIP (Supplementary Table 4A) is LDL particle diameter (LDL.D, MIP = 0.526). All other risk factors have evidence less than MIP < 0.25. In order to check the model fit, we consider the best individual models (Supplementary Table 4B) with posterior probability >0.02. For illustration, we present here the predicted associations with AMD based on the best model including LDL.D, and TG content in small HDL (S.HDL.TG) against the observed associations with AMD. We colour code genetic variants according to their $q$-statistic (Fig. 4a and Supplementary Fig. 16A, Supplementary Table 5) and Cook's distance (Fig. 4b and Supplementary Fig. 16B, Supplementary Table 6). First, the $q$-statistic indicates two variants, rs492602 in the *FUT2* gene region and rs6859 in the *APOE* gene region, as outliers in all best models. Second, the genetic variant with the largest Cook's distance ($Cd = 0.871$–$1.087$) consistently in all models investigated is rs261342 mapping to the *LIPC* gene region. This variant has been indicated previously to have inconsistent associations with AMD compared with other genetic variants[23,24].

We repeat the analysis without the three influential and/or heterogeneous variants ($n = 145$), and report the ten risk factors with the largest marginal inclusion probability in Table 2 and the full results in Supplementary Table 7. The top two risk factors are total cholesterol in extra-large HDL particles (XL.HDL.C, MIP = 0.700) and total cholesterol in large HDL particles (L.HDL.C, MIP = 0.229). XL.HDL.C and L.HDL.C were strongly correlated ($r = 0.80$), and models including both have very low evidence as can be seen in Table 3 which gives the posterior probability of individual models. Supplementary Fig. 17 shows the scatterplots of the genetic associations with each of these two risk factors individually against the genetic associations with

AMD risk. We select the five individual models with a posterior probability >0.02 to inspect the model fit (Supplementary Figs. 18 and 19). This time, no genetic variant has a consistently large $q$-statistic (Supplementary Table 8) or Cook's distance (Supplementary Table 9). Repeating the analysis without the largest influential point, rs5880 in the *CETP* gene region, or the strongest outlier, rs103294 in the *AC245884.7* gene region, did not impact the ranking of the risk factors. We tested the robustness of the results with respect to a wide range of prior variance and prior probability parameters; results did not change substantially (Supplementary Tables 10 and 11).

We also applied Lars, Lasso and Elastic Net after excluding outliers and influential points ($n = 145$). Lars showed the largest regression coefficient for L.HDL.C including 11 risk factors. Lasso selected four risk factors with the largest regression coefficient for XL.HDL.C, while Elastic Net selected ten risk factors with the largest regression coefficient for L.HDL.C. Full results for the competing methods are given in Supplementary Tables 12–15. A disadvantage of regularised regression approaches is that risk factor selection is binary; risk factors are either included in the model or set to have a coefficient zero. The magnitude of regularised regression coefficients does not rank risk factors according to their strength of evidence for inclusion in the model.

The detection of influential points in the initial analysis highlights rs26134, a genetic variant in the *LIPC* gene region, which had a strong impact on the analysis. Figure 5 shows the model diagnostics of the highest ranked model excluding outlying and influential points (XL.HDL.C as the sole risk factor), with the variant in the *LIPC* gene region also plotted. This particular variant exhibits a distinct, potentially pleiotropic, effect. While all other variants support that XL.HDL.C increases the risk of AMD, this particular variant has the opposite direction of association with AMD risk as that predicted by its association with XL.HDL.C. Further functional and fine-mapping studies of this region are needed to understand the contrasting association of this variant with AMD risk.

These results confirm previous studies[23,24] that identified HDL cholesterol as a putative risk factor for AMD and draw the attention to extra-large and large HDL particles. A recent observational study[22] supports our finding that extra-large HDL particles have an important role in the pathogenesis of AMD.

## Discussion

We here introduce MR-BMA, an approach for multivariable MR which allows for the analysis of high-throughput experiments. This model averaging procedure prioritises and selects causal risk factors in a Bayesian framework from a high-dimensional set of related candidate risk factors. As is common for statistical techniques for variable selection, MR-BMA does not provide unbiased estimates. However, as shown in the simulation study, causal estimates from MR-BMA have reduced variance and thus MR-BMA improves over unbiased approaches, like the IVW method, in terms of mean squared error and detection of true risk factors. The primary aim of this work is to detect causal risk factors rather than to unbiasedly estimate the magnitude of their causal effects. MR-BMA is a multivariable MR approach that can analyse a high-dimensional set of risk factors. When analysing many risk factors jointly one important implicit assumption of MR-BMA is sparsity, i.e., the proportion of true causal risk factors compared with all risk factors considered is small. Since MR-BMA evaluates all possible combinations of risk factors exhaustively or all relevant combinations of risk factors in a shotgun stochastic search there is an upper bound for the maximum model size in order to keep the computation tractable. Sparsity is a common assumption for high-throughput data and we have seen in the applied example that the best models only

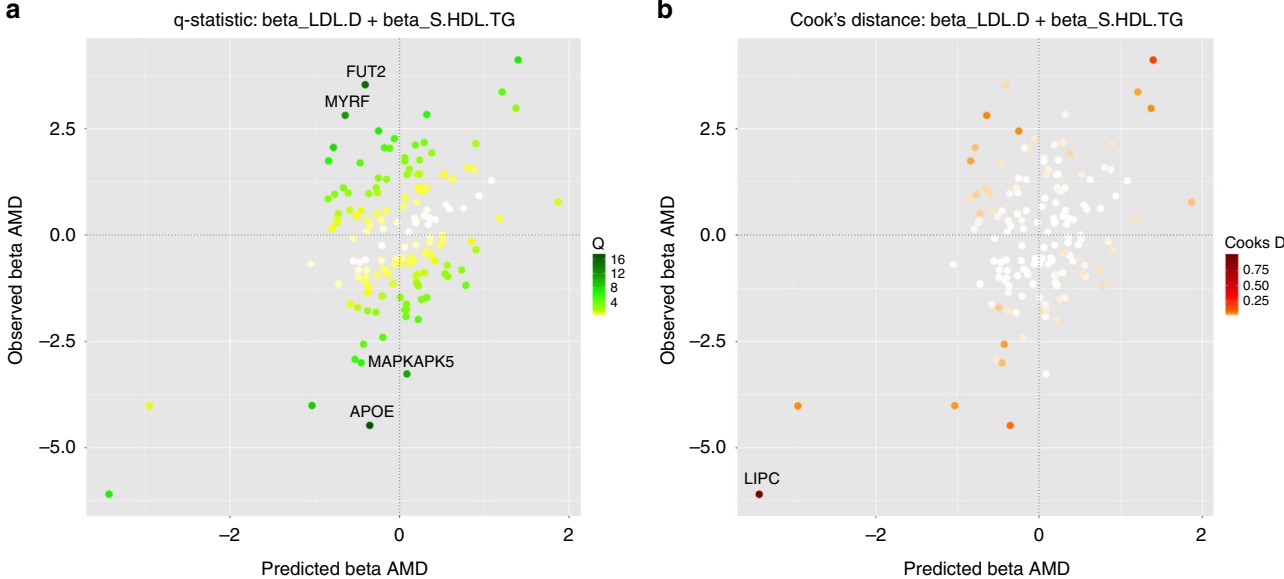

**Fig. 4 Diagnostic plots for outliers and influential genetic variants.** Plotting the predicted associations with AMD based on the model including LDL.D, and S.HDL.TG (x-axis) against the observed associations with AMD (y-axis) showing all $n = 148$ genetic variants. This is the highest-ranking model when keeping outlying and influential genetic variants in the analysis. The colour code shows: **a** the q-statistic for outliers and **b** Cook's distance for the influential points. Any genetic variant with q-value larger than 10 or Cook's distance larger than the median of the relevant $F$-distribution is marked by a label indicating the gene region.

**Table 2 Ranking of risk factors for age-related macular degeneration (AMD) according to their marginal inclusion probability (MIP) after exclusion of outlying and influential variants ($n = 145$).**

|   | Risk factor | Marginal inclusion probability (MIP) | Model-averaged causal effect $\hat{\theta}_{MACE}$ |
|---|---|---|---|
| 1 | XL.HDL.C | 0.7 | 0.344 |
| 2 | L.HDL.C | 0.229 | 0.087 |
| 3 | HDL.D | 0.087 | 0.022 |
| 4 | XS.VLDL.TG | 0.082 | −0.019 |
| 5 | LDL.D | 0.074 | −0.018 |
| 6 | IDL.TG | 0.066 | −0.012 |
| 7 | XXL.VLDL.TG | 0.063 | 0.018 |
| 8 | S.VLDL.TG | 0.062 | −0.014 |
| 9 | Serum.TG | 0.061 | −0.014 |
| 10 | Serum.C | 0.054 | −0.011 |

Results are given after excluding genetic variants in the *APOE*, *FUTC*, and *LIPC* regions. $\hat{\theta}_{MACE}$ is the model-averaged causal effect of a risk factor
HDL.D HDL diameter, IDL.TG triglycerides in IDL, L.HDL.C total cholesterol in large HDL, LDL.D LDL diameter, Serum.C serum total cholesterol, Serum.TG serum total triglycerides, S.VLDL.C total cholesterol in small VLDL, S.VLDL.TG triglycerides in small VLDL, XS.VLDL.TG triglycerides in very small VLDL, XL.HDL.C total cholesterol in very large HDL

contained one to three metabolites as risk factors despite allowing for a model size of up to twelve risk factors. Yet this is an important aspect of the algorithm and the maximum model size should be adjusted if models including many risk factors are expected or evidenced in the data.

We demonstrated the approach with application to a dataset of NMR metabolites, which included predominantly lipid measurements, using variants associated with lipids as instrumental variables. Previous MR analysis[23,24] including three lipid measurements from the Global Lipids Genetics Consortium[19] have identified HDL cholesterol as potential risk factor for AMD. Our approach to multivariable MR refined this analysis and confirmed HDL cholesterol as a potential causal risk factor for AMD, further

pinpointing that large or extra-large HDL particles are likely to be driving disease risk. Other areas of application where this method could be used include imaging measurements of the heart and coronary artery disease, body composition measures and type 2 diabetes, or blood cell traits and atherosclerosis. As multivariable MR accounts for measured pleiotropy, this approach facilitates the selection of suitable genetic variants for causal analyses. In each case, it is likely that genetic predictors of the set of risk factors can be found, even though finding specific predictors of, for example, particular heart measurements from cardiac imaging, may be difficult given widespread pleiotropy[25]. MR-BMA allows a more agnostic and hypothesis-free approach to causal inference, allowing the data to identify the causal risk factors.

Multivariable MR estimates the direct effect of a risk factor on the outcome and not the total effect as estimated in standard univariable MR. This is in analogy with multivariable regression where the regression coefficients represent the association of each variable with the outcome when all others are held constant. Having said this, the main goal of our approach is risk factor selection, and not the precise estimation of causal effects, since the variable selection procedure shrinks estimates towards the null. This results in causal effect estimates being biased towards the Null when there is a causal effect and unbiased estimates when there is no causal effect. If there are mediating effects between the risk factors, then this approach will identify the risk factor most proximal to and has the most direct effect on an outcome. For example, if the risk factors included would form a signalling cascade (Supplementary Fig. 1b) then our approach would identify the downstream risk factor in the cascade with the direct effect on the outcome and not the upstream risk factors in the beginning of the cascade. Hence, a risk factor may be a cause of the outcome, but if its causal effect is mediated via another risk factor included in the analysis, then it will not be selected in the multivariable MR approach.

Our approach is formulated in a Bayesian framework. Particular care needs to be taken when choosing the hyper-parameter for the prior probability which relates to the a priori expected number of causal risk factors. In the applied example the results

| | Models or sets of risk factor(s) | Posterior probability (PP) | Model-specific causal estimates $\hat{\theta}_y$ |
|---|---|---|---|
| 1 | XL.HDL.C | 0.156 | 0.509 |
| 2 | L.HDL.C | 0.078 | 0.384 |
| 3 | XL.HDL.C,XS.VLDL.TG | 0.026 | 0.457,−0.181 |
| 4 | IDL.TG,XL.HDL.C | 0.025 | −0.179,0.495 |
| 5 | HDL.D | 0.023 | 0.359 |
| 6 | Serum.C,XL.HDL.C | 0.019 | −0.183,0.573 |
| 7 | S.VLDL.TG,XL.HDL.C | 0.015 | −0.172,0.443 |
| 8 | S.VLDL.C,XL.HDL.C | 0.014 | −0.164,0.477 |
| 9 | Serum.TG,XL.HDL.C | 0.014 | −0.169,0.465 |
| 10 | S.HDL.TG,XL.HDL.C | 0.013 | −0.18,0.415 |

**Table 3 Ranking of models (sets of risk factors) for age-related macular degeneration (AMD) according to their posterior probability (PP) after exclusion of outlying and influential variants ($n = 145$).**

Results are given after excluding genetic variants in the *APOE*, *FUTC*, and *LIPC* regions. $\hat{\theta}_y$ is the causal effect estimate for a specific model
HDL.D HDL diameter, IDL.TG triglycerides in IDL, L.HDL.C total cholesterol in large HDL, LDL.D LDL diameter, Serum.C serum total cholesterol, Serum.TG serum total triglycerides, S.VLDL.C total cholesterol in small VLDL, S.VLDL.TG triglycerides in small VLDL, XS.VLDL.TG triglycerides in very small VLDL, XL.HDL.C total cholesterol in very large HDL

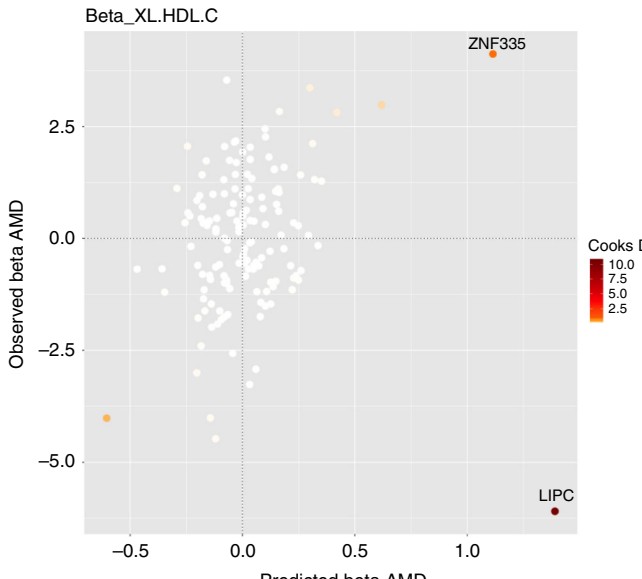

**Fig. 5 Diagnostic plot for influential genetic variants.** Plotting the predicted associations with AMD based on the model including XL.HDL.C (*x*-axis) against the observed associations with AMD (*y*-axis) showing all $n = 148$ genetic variants, where the colour code shows Cook's distance for the genetic variants. This is the highest-ranking model on omission of outlying and influential genetic variants from the analysis. Note rs26134 in the *LIPC* gene region which has an anomalous direction of association with AMD risk in contrast to all other genetic variants.

were robust to a wide range of prior specifications for the parameter as seen in Supplementary Table 10. In addition, the prior variance of the causal parameters needs to be specified and tested for robustness as we show in the Supplementary Table 11.

When genetic variants are weak predictors for the risk factors, this can introduce weak instrument bias. In univariable two-sample MR, any bias due to weak instruments is towards the null and does not lead to inflated type 1 error rates[26]. However, in multivariable MR, weak instrument bias can be in any direction (see Methods), although bias will tend to zero as the sample size increases and consequently the instrument strength increases. Selection of risk factors is only possible if there are genetic variants that are predictors of these risk factors. One of the biggest challenges of multivariable MR is the design of a meaningful

study, in particular the choice of both, the genetic variants and the risk factors. The design of the study is important for the interpretation of the risk factors prioritised: The ranking of risk factors is conditional on the genetic variants used. For instance, in our applied example we find evidence for extra-large and large HDL cholesterol concentration given that we used lipid-related genetic variants as instrumental variables. We recommend to include only risk factors which have at least one, and ideally multiple genetic variants that act as strong instruments. Caution is needed for the interpretation of null findings, particularly in our example for non-lipid risk factors, as these might be deprioritised in terms of statistical power by our choice of genetic variants. The instrument selection and general study design are essential for the MR-BMA approach and we strongly recommend the user to be critical in the choice of genetic variants and risk factors. Moreover, similar to standard MR we urge to perform model checks and be transparent in the presentation of the removal of outlier/influential genetic variants.

A further requirement for multivariable MR is that the genetic variants can distinguish between risk factors[12]. We recommend to check the correlation structure between genetic associations for the selected genetic variants and to include no pair of risk factors which is extremely strongly correlated. In the applied example, we included only risk factors with an absolute correlation <0.99. As we were not able to include more than three measurements for each lipoprotein category (cholesterol content, triglyceride content, diameter), care should be taken not to overinterpret findings in terms of the specific measurements included in the analysis rather than those correlated measures that were excluded from the analysis (such as phospholipid and cholesterol ester content).

Another assumption of multivariable MR is that there is no unmeasured horizontal pleiotropy. This means that the variants do not influence the outcome except via the measured risk factors. The assumption of no horizontal pleiotropy is a common and untestable assumption in MR. It is an active area of research to robustify MR against violations of this assumption. Some of these robust methods for MR make a specific assumption about the behaviour of pleiotropic variants, such as MR-Egger[27], which assumes pleiotropic effects are uncorrelated from the genetic associations with the risk factor—the InSIDE assumption. Other methods exclude outlying variants as they are potentially pleiotropic such as MR-PRESSO[28]. In multivariable MR, pleiotropic variants can be detected as outliers to the model fit. Here we quantify outliers using the *q*-statistic. Outlier detection in standard univariable MR can be performed by model averaging where different subsets of instruments are considered[29,30], assuming

that a majority of instruments is valid, but without prior knowledge which are the valid instruments. In multivariable MR, ideally one would like to perform model selection and outlier detection simultaneously. In addition, we search for genetic variants that are influential points. While these may not necessary be pleiotropic, we suggest removing such variants as a sensitivity analysis to judge whether the overall findings from the approach are dominated by a single variant. Findings are likely to be more reliable when they are evidenced by multiple genetic variants. One necessary future development is post-selection inference[31,32] in the high-dimensional multivariable MR framework. MR-BMA does not provide unbiased causal effect estimates. Re-fitting an unbiased multivariable MR model after risk factor selection would ignore the uncertainty of the selection and consequently not provide valid inferences.

In conclusion, we introduce here MR-BMA, the first approach to perform risk factor selection in multivariable MR, which can identify causal risk factors from a high-throughput experiment. MR-BMA can be used to determine which out of a set of related risk factors with common genetic predictors are the causal drivers of disease risk.

## Methods

**Mendelian randomization data input: summarized data set-up.** One of the key features of MR is that the approach can be performed using summarised data on genetic associations—beta-coefficients and their standard errors from univariate regression analyses. No access to individual-level genotype data is needed. In addition, these association estimates can be derived from different samples. In two-sample MR, the genetic associations with the risk factor are derived from one sample and the genetic associations with the outcome from another sample[26]. The use of summarised data in two-sample MR allows the sample size to be maximised by integrating data from large meta-analyses including hundreds of thousands of participants.

We assume the context of two-sample MR with summarized data[33]. For each genetic variant $i = 1, \ldots, n$ and each risk factor $j = 1, \ldots, d$, we take the beta-coefficient $\beta^\star_{X_{ij}}$ and standard error $\text{se}(\beta^\star_{X_{ij}})$ from a univariable regression in which the risk factor $\mathbf{X}_j$ is regressed on the genetic variant $\mathbf{G}_i$ in sample one, and beta-coefficient $\beta^\star_{Y_i}$ and standard error $\text{se}(\beta^\star_{Y_i})$ from a univariable regression in which the outcome $\mathbf{Y}$ is regressed on the genetic variant $\mathbf{G}_i$ in sample two. For simplicity of notation, although the beta-coefficients are estimates, we omit the conventional "hat" notation and treat the beta-coefficients as observed data points. When considering multiple risk factors, we construct a matrix of beta-coefficients $\boldsymbol{\beta}^\star_X$ of dimension $n \times d$, where $d$ is the number of risk factors and $n$ is the number of genetic variants.

We assume that the genetic effects on risk factors and on the outcome are linear and homogeneous across the population, and identical between the two samples[34]. Furthermore, we assume that the $n$ genetic variants selected as instrumental variables are independent, an assumption common in MR studies. This is usually achieved by including only the lead genetic variant from each gene region in the analysis. Finally, we assume that genetic association estimates are derived from two distinct samples with no overlap between the samples. These assumptions can all be relaxed to some extent if the goal is causal inference rather than causal estimation; see ref. [35] for details.

**Multivariable Mendelian randomization and the linear model.** Multivariable MR is an extension of the standard MR paradigm (Fig. 1) to model not one, but multiple risk factors as illustrated in Fig. 2. Univariable MR can be cast as a weighted linear regression model in which the genetic associations with the outcome $\beta^\star_{Y_i}$ are regressed on the genetic associations with the risk factor $\beta^\star_{X_i}$ in order to estimate the total effect $\theta$ of the risk factor $\mathbf{X}$ on the outcome $\mathbf{Y}$[36]

$$\beta^\star_{Y_i} = \theta \beta^\star_{X_i} + \epsilon_i, \quad \epsilon_i \sim \mathcal{N}(0, \text{se}(\beta^\star_{Y_i})^2). \tag{1}$$

In multivariable MR, the genetic associations with the outcome are regressed on the genetic associations with all the $j = 1, \ldots, d$ risk factors[10]

$$\beta^\star_{Y_i} = \theta_1 \beta^\star_{X_{i1}} + \theta_2 \beta^\star_{X_{i2}} + \ldots + \theta_d \beta^\star_{X_{id}} + \epsilon_i, \quad \epsilon_i \sim \mathcal{N}(0, \text{se}(\beta^\star_{Y_i})^2). \tag{2}$$

Weights in these regression models are proportional to inverse of the variance of the genetic association with the outcome $(\text{se}(\beta^\star_{Y_i})^{-2})$. This is to ensure that genetic variants having more precise association estimates receive more weight in the analysis. The same weighting can also be achieved by standardising the association estimates, by dividing $\beta^\star_{Y_i}$ and $\beta^\star_{X_i}$ by $\text{se}(\beta^\star_{Y_i})$. In the following derivations, we assume that $\beta_{Y_i} = \beta^\star_{Y_i}/\text{se}(\beta^\star_{Y_i})$ and $\beta_{X_i} = \beta^\star_{X_i}/\text{se}(\beta^\star_{Y_i})$ are standardised, so that the variances of the $\epsilon_i$ terms are all 1. To account for heterogeneity in the regression equation, we can use a multiplicative random effects model, which increases the variance of the error terms by a multiplicative factor[37].

Our parameter of interest is the vector of regression coefficients $\boldsymbol{\theta} = \{\theta_1, \ldots, \theta_d\}$. These are the direct causal effects of the risk factors in turn on the outcome when all the other risk factors in the model are held constant[13]. In contrast, univariable Mendelian randomization using genetic variants that are instrumental variables for the specific risk factor of interest estimates the total effect of the risk factor on the outcome. The direct effect will differ from the total effect if the effect of risk factor is mediated via another risk factor included in the model[12]. We illustrate the difference between the direct and total effect using directed acyclic graphs in Supplementary Fig. 1. In some cases (such as to identify the proximal risk factor to the outcome), the direct effect is of interest; in other cases (such as to evaluate the potential impact of intervening on a risk factor), it is the total effect that is truly of interest[13].

**Choosing genetic variants as instruments.** In multivariable MR, a genetic variant is a valid instrumental variable if the following criteria hold:

IV1 Relevance: The variant is associated with at least one of the risk factors.
IV2 Exchangeability: The variant is independent of all confounders of each of the risk factor–outcome associations.
IV3 Exclusion restriction: The variant is independent of the outcome conditional on the risk factors and confounders.

One of the main differences of multivariable MR compared with univariable MR is the relaxation of the exclusion restriction condition. In contrast to univariable MR, multivariable MR allows for measured pleiotropy[14] via any of the observed risk factors. Hence the instrumental variable assumptions are more likely to be satisfied for multivariable MR than for univariable MR for a given choice of genetic variants.

It is not necessary for every genetic variant to be associated with all the risk factors, although if no genetic variants are associated with a particular risk factor, then the causal effect of that risk factor cannot be estimated. This would also occur if the genetic associations with two risk factors were exactly proportional. For precise identification of causal risk factors, it is necessary to have some variants that are more strongly associated with particular risk factors than others[12]. More precisely a risk factor can be included into the analysis if the following criteria (RF1–RF2) hold:

RF1 Relevance: The risk factor needs to be strongly instrumented by at least one genetic variant included as instrumental variable.
RF2 No multi-collinearity: The genetic associations of any risk factor included cannot be linearly explained by the genetic associations of any other risk factor or by the combination of genetic associations of multiple other risk factors included in the analysis.

The study design and in particular the selection of genetic variants as IVs are the most important steps for multivariable MR and great care needs to be taken when designing the study and also when reporting the study design. All interpretation of the results is conditional on the genetic variants selected as IVs. We initially assume that all genetic variants are valid instruments. There is an emerging literature[27,38] on how to perform robust MR analysis in the presence of invalid instruments; similar extensions can be adapted for multivariable MR[14].

**Risk factor selection as variable selection in the linear model.** We consider the situation in which we have a set of genetic variants that are instrumental variables for a set of risk factors, and we want to select which of those risk factors are causes of the outcome. Our implicit prior belief is that not all of the risk factors are causally related to the outcome and that there are some true causal risk factors (signal) and some risk factors which do not have an effect (noise). We formulate the selection of risk factors in two-sample multivariable MR as a variable selection task in the linear regression framework. In order to model the correlation between risk factors we base our likelihood on a Gaussian distribution

$$\boldsymbol{\beta}_{\mathbf{Y}} | \boldsymbol{\beta}_{\mathbf{X}}, \boldsymbol{\theta}, \tau \sim N\left(\boldsymbol{\beta}_{\mathbf{X}}\boldsymbol{\theta}, \frac{1}{\tau}\right). \tag{3}$$

Following the $D_2$ prior specifications as introduced in ref. [39], we use the following conjugate priors for the causal effects $\boldsymbol{\theta}$, the residual error $\epsilon$, and the precision $\tau$

$$\boldsymbol{\theta} \sim N(0, \boldsymbol{\nu}/\tau)$$
$$\epsilon \sim N\left(0, \frac{1}{\tau}\right) \tag{4}$$
$$\tau \sim \Gamma(\kappa/2, \lambda/2),$$

where $\boldsymbol{\nu} = \text{diag}(\sigma^2)$ is the diagonal variance matrix of the causal effects (independence prior), and the precision $\tau$ is assumed to follow a Gamma distribution with hyperparameters $\kappa$ as the shape and $\lambda$ as the scale parameter. Next, we introduce a binary indicator $\boldsymbol{\gamma}$ of length $d$ that indicates which risk factors are selected and which ones are not

$$\gamma_j = \begin{cases} 1, & \text{if the } j\text{th risk factor is selected,} \\ 0 & \text{otherwise.} \end{cases} \tag{5}$$

The indicator $\boldsymbol{\gamma}$ encodes a specific regression model $M_{\boldsymbol{\gamma}}$ that includes the risk factors as indicated in $\boldsymbol{\gamma}$. A model $M_{\boldsymbol{\gamma}}$ can include one or a combination of multiple risk

factors. To evaluate the evidence of a specific model $M_\gamma$, we calculate the Bayes factor for model $M_\gamma$ against the null model that does not include an intercept or any risk factor. The Bayes factor $\mathrm{BF}(M_\gamma)$ has the following closed form representation

$$\mathrm{BF}(M_\gamma) = \frac{|\mathbf{\Omega}|^{1/2}}{|\mathbf{v}_\gamma|^{1/2}} \left( \frac{\boldsymbol{\beta}_{\mathbf{Y}}^t \boldsymbol{\beta}_{\mathbf{Y}} - \mathbf{\Theta}^t \mathbf{\Omega}^{-1} \mathbf{\Theta}}{\boldsymbol{\beta}_{\mathbf{Y}}^t \boldsymbol{\beta}_{\mathbf{Y}}} \right)^{-n/2}, \qquad (6)$$

where $\mathbf{\Theta} = \mathbf{\Omega} \boldsymbol{\beta}_{\mathbf{X}_\gamma}^t \boldsymbol{\beta}_{\mathbf{Y}}$ is the causal effect estimate and $\mathbf{\Omega} = (\mathbf{v}_\gamma^{-1} + \boldsymbol{\beta}_{\mathbf{X}_\gamma}^t \boldsymbol{\beta}_{\mathbf{X}_\gamma})^{-1}$ is the inverse of the shrinkage covariance between the genetic associations of the risk factors. For a detailed derivation of the Bayes factor we refer to the Supplementary Methods in the Supplementary Information.

**Prior specification.** Another important aspect is the prior for the model size $k$, which we model using a Binomial distribution

$$Pr(K = k) = \binom{d}{k} p^k (1-p)^{d-k}. \qquad (7)$$

This requires choosing the probability $p$ of including a risk factor in the model according to prior assumptions regarding the sparsity of the results. We recommend to select $p$ according to the expected a priori model size, which is $p \times d$. Currently, all risk factors are assumed to have the same prior probability, and thus the probability of all models of the same size $k$ is equal. The prior of a specific model $M_\gamma$ of size $k$ is defined as

$$p(M_\gamma) = \binom{d}{k}^{-1} Pr(K = k) = p^k (1-p)^{d-k}. \qquad (8)$$

The second important aspect is the prior for the variance of the risk factors $\mathbf{v} = \mathrm{diag}(\sigma^2)$, where we assume that all risk factors have the same prior variance $\sigma^2$. Large values of $\sigma^2$ would favour strong causal effects of the risk factors on the outcome. Following ref. [39] we initially set $\sigma^2 = 0.25$, but sensitivity of the results with respect to this prior should be investigated. The parameter can be specified in the implementation of MR-BMA. In the applied example we perform a sensitivity analysis for this important parameter.

**Posterior calculation and marginal inclusion probability of a risk factor.** Let $\mathbf{\Gamma}$ be the space of all possible combinations of risk factors. The posterior probability (PP) of a model $M_\gamma$ can be expressed by the prior probability (8) and the Bayes factor (6) of model $M_\gamma$ is

$$\mathrm{PP}(M_\gamma | \boldsymbol{\beta}_{\mathbf{Y}}, \boldsymbol{\beta}_{\mathbf{X}}) = \frac{p(M_\gamma) \mathrm{BF}(M_\gamma)}{\sum_{\gamma \in \Gamma} p(M_\gamma) \mathrm{BF}(M_\gamma)}. \qquad (9)$$

In high-dimensional variable selection, the evidence for one particular model can be small because the model space is very large and many models might have comparable evidence. This is why MR-BMA uses Bayesian model averaging (BMA) and computes for each risk factor $j$ its marginal inclusion probability (MIP), which is defined as the sum of the posterior probabilities over all models where the risk factor is present

$$\mathrm{MIP}(j = 1 | \boldsymbol{\beta}_{\mathbf{Y}}, \boldsymbol{\beta}_{\mathbf{X}}) = \frac{\sum_{\gamma \in \Gamma} I(\gamma_j = 1) p(M_\gamma) \mathrm{BF}(M_\gamma)}{\sum_{\gamma \in \Gamma} p(M_\gamma) \mathrm{BF}(M_\gamma)}, \qquad (10)$$

where $I(\gamma_j = 1)$ equals 1 if risk factor $j$ is part of the model and 0 otherwise.

An exhaustive evaluation of all possible combinations of risk factors is computationally prohibitive already for a moderate number of risk factors ($d > 20$). To alleviate this issue we have implemented a shotgun stochastic search algorithm[40] that evaluates all combinations of risk factors with a non-negligible contribution to the calibration factor $\sum_{\gamma \in \Gamma} P(M_\gamma) \mathrm{BF}(M_\gamma)$ in Eq. (9). This algorithm is based on the assumption that the majority of combinations of risk factors have a posterior probability close to zero and do not need to be considered when computing the calibration factor in the denominator of Eqs. (9) and (10).

**Causal estimation.** We derive the estimates for the causal effects $\hat{\boldsymbol{\theta}}_\gamma$ of model $M_\gamma$ as

$$\hat{\boldsymbol{\theta}}_\gamma = \mathbf{\Omega} \boldsymbol{\beta}_{\mathbf{X}_\gamma}^t \boldsymbol{\beta}_{\mathbf{Y}} = (\mathbf{v}_\gamma^{-1} + \boldsymbol{\beta}_{\mathbf{X}_\gamma}^t \boldsymbol{\beta}_{\mathbf{X}_\gamma})^{-1} \boldsymbol{\beta}_{\mathbf{X}_\gamma}^t \boldsymbol{\beta}_{\mathbf{Y}}, \qquad (11)$$

which is closely related to the regression coefficient in Ridge regression. Adding the diagonal matrix $\mathbf{v}_\gamma^{-1}$ stabilises the inversion and makes the estimate more robust to strong correlation among risk factors. There can be strong correlation between candidate risk factors as seen in the genetic correlation matrices in the applied examples as illustrated in Supplementary Figs. 2 and 11, which makes it important to stabilise the causal estimate.

The model-averaged causal estimate (MACE) for risk factor $j$ from the MR-BMA approach is

$$\hat{\boldsymbol{\theta}}_{\mathrm{MACE}}(j) = \sum_{\gamma \in \Gamma} I(\gamma_j = 1) \mathrm{PP}(M_\gamma | \boldsymbol{\beta}_{\mathbf{Y}}, \boldsymbol{\beta}_{\mathbf{X}}) \hat{\boldsymbol{\theta}}_\gamma. \qquad (12)$$

Both $\hat{\boldsymbol{\theta}}_\gamma$ and $\hat{\boldsymbol{\theta}}_{\mathrm{MACE}}$ are conservative estimates of the true causal effect. They are biased towards the Null if there is a causal effect and unbiased otherwise. We therefore only recommend to interpret the direction of effect and the magnitude of these causal effect estimates in comparison with the effects of other risk factors.

MR-BMA ranks and prioritises risk factors according to their marginal inclusion probability and estimates the MACE as defined in Eq. (12). As an alternative approach, we also consider selecting the 'best model' based on the individual model posterior probabilities as defined in Eq. (9).

**Detection of invalid and influential instruments.** Invalid instruments may be detected as outliers with respect to the fit of a specific linear model $M_\gamma$. We recommend to check the best individual models for outliers by visual inspection of the scatterplot of the predicted associations based on $M_\gamma$ with the outcome $\hat{\boldsymbol{\beta}}_{\mathbf{Y}} = \boldsymbol{\beta}_{\mathbf{X}_\gamma} \hat{\boldsymbol{\theta}}_\gamma$ against the actual observed $\boldsymbol{\beta}_{\mathbf{Y}}$. If a genetic variant is detected consistently as an outlier in several of the top models, it may be advisable to explore the analyses excluding that outlying variant from the analysis. To quantify outliers we use the $Q$-statistic, which is an established tool for identifying heterogeneity in meta-analysis[15]. It is defined as the sum of the residual vector $q$, which is the squared difference between the observed and predicted association with the outcome

$$Q = \sum_{i=1}^{n} q_i = \sum_{i=1}^{n} (\beta_{Y_i} - \hat{\beta}_{Y_i})^2. \qquad (13)$$

We note that Eq. (13) is defined on the weighted coefficients $\beta_{Y_i}$. When considering the unweighted coefficients $\beta_{Y_i}^\star$ the $Q$-statistic[12] is defined as

$$Q = \sum_{i=1}^{n} q_i = \sum_{i=1}^{n} \frac{1}{\mathrm{se}(\beta_{Y_i}^\star)^2} (\beta_{Y_i}^\star - \hat{\beta}_{Y_i}^\star)^2, \qquad (14)$$

with first order weighting equal to $\frac{1}{\mathrm{se}(\beta_{Y_i})^2}$[41].

The individual element $q_i$ measures the heterogeneity of genetic variant $i$ for a particular model $M_\gamma$. We refer to $q_i$ as the q-statistic, and use this to evaluate if specific genetic variants are outliers to the model fit.

Even if there are no outliers, it is advisable to check for influential observations and re-run the approach omitting a particular influential variant from the analysis. If a particular genetic variant has a strong association with the outcome, then it may have undue influence on the variable selection, leading to a model that fits that particular observation well, but other observations poorly. To quantify influential observations for a particular model $M_\gamma$ we suggest to use Cook's distance[16]

$$Cd_i = \frac{q_i}{s^2 d} \frac{h_i}{(1-h_i)^2}, \qquad (15)$$

where $h_i$ is the $i$th diagonal element of the hat matrix $\mathbf{H} = \boldsymbol{\beta}_{\mathbf{X}_\gamma} (\mathbf{v}_\gamma^{-1} + \boldsymbol{\beta}_{\mathbf{X}_\gamma}^t \boldsymbol{\beta}_{\mathbf{X}_\gamma})^{-1} \boldsymbol{\beta}_{\mathbf{X}_\gamma}^t$, and $s^2 = \frac{1}{n-d} \epsilon^t \epsilon$ is the mean squared error of the regression model. Following ref. [42], we recommend to use the median of a central $F$-distribution with $d$ and $n - d$ degrees of freedom as a threshold, and remove variants that have a Cook's distance which exceeds this value.

**Impact of weak instrument bias.** In the following presentation, we consider two risk factors with observed genetic associations $\beta_{X_1}$ and $\beta_{X_2}$, which are a sum of the true genetic associations $\beta_{X_1}^\dagger$ and $\beta_{X_2}^\dagger$ and an additional error term $\epsilon_1$ and $\epsilon_2$, respectively, i.e.

$$\beta_{X_1} = \beta_{X_1}^\dagger + \epsilon_1$$
$$\beta_{X_2} = \beta_{X_2}^\dagger + \epsilon_2.$$

From this we define $\lambda_1 = \frac{\mathrm{var}(\epsilon_1)}{\mathrm{var}(\beta_{X_1}^\dagger)}$ as the ratio of the uncertainty in the estimates of the genetic associations ($\mathrm{var}(\epsilon_1)$) over the variability of the true genetic associations $\mathrm{var}(\beta_{X_1}^\dagger)$, and we define $\lambda_2$ similarly. Further let $\rho$ be the correlation between $\beta_{X_1}$ and $\beta_{X_2}$, and let $\theta_1$ and $\theta_2$ be the true direct effect of $X_1$ on $Y$ and $X_2$ on $Y$, respectively. Following the measurement error literature[43], we derive the induced bias of the IVW estimates of the true causal effects $\theta_1$ and $\theta_2$, respectively, as

$$\hat{\theta}_1 = \theta_1 - \frac{\theta_1 \lambda_1 - \rho \theta_2 \lambda_2}{1 - \rho}$$
$$\hat{\theta}_2 = \theta_2 - \frac{\theta_2 \lambda_2 - \rho \theta_1 \lambda_1}{1 - \rho},$$

where $\hat{\theta}_1$ and $\hat{\theta}_2$ are the expected values of the effects for the mismeasured genetic association estimates. Looking closer at $\lambda$, the variability across variants of the true genetic associations, $\mathrm{var}(\beta_X^\dagger)$, is related to instrument strength. Thus the induced bias will be smaller the stronger the instruments. At the same time the uncertainty of the genetic association estimates, $\mathrm{var}(\epsilon)$, decreases when increasing the sample size. If the genetic associations with the risk factors are estimated with different degrees of uncertainty, then bias could be more considerable. Analogous to differential measurement error, risk factors with more precisely estimated genetic associations would be prioritized in the regression model. In our application, all risk factors are measured

on the same high-throughput platform and on the same sample size, thus reducing the impact of weak instrument bias to influence the ranking of risk factors.

**Simulation study.** To evaluate the performance of MR-BMA, we perform a simulation study taking genetic associations with risk factors from two real data sets, the first one based on genetic associations with NMR metabolites[11] and secondly on genetic associations with blood cell traits[4]. Further information on the data sets and pre-processing is given in the next sections. We simulate genetic associations with the outcome $\beta_Y$ based on a subset of risk factors selected at random, which we refer to as the 'true' risk factors. We investigate three different scenarios and six sets of parameter values per scenario:

Size of the data set: small ($d = 12$ NMR metabolites selected at random), large ($d = 92$ all NMR metabolites available), and moderate ($d = 33$ all blood cell traits available) number of risk factors included.
Number of true risk factors: (Setting A) four risk factors have an effect of $\theta = 0.3$, the other risk factors have no effect; (Setting B) four risk factors have an effect of $\theta = 0.3$, and another four risk factors have an effect of $\theta = -0.3$, the other risk factors have no effect.
Proportion of variance in the outcome explained by the risk factors: $R^2 = 0.1, 0.3, 0.5$ which defines the variance of the error.

We compare six different analysis methods:

Multivariable inverse-variance weighted (IVW) regression (Eq. (2))[10]
Least-angle regression (Lars) as L1 regularised regression[17]
Lasso as L1 regularised regression[18]
Elastic Net as L1 and L2 regularised regression[18]
MR-BMA using marginal inclusion probabilities (Eq. (10))
Bayesian best model selection using posterior probabilities of individual models (Eq. (9))

Both Lars[17] and Lasso[18] are versions of L1 regularised linear regression, and Elastic Net is a mixture of a L1 and L2 regularised linear regression, all of which have been devised for variable selection in high-dimensional data. We use here the Lars implementation[17] and for Lasso and Elastic Net we use the glmnet[18] implementation. For all regularised regression methods, we use cross-validation (CV) to tune the regularisation parameter to achieve the minimum cross-validation MSE. For the small risk factor space including 12 NMR metabolites, the MR-BMA approach is performed using an exhaustive search of all possible models with prior probability of a risk factor to be included set to $p = 0.5$, while for the moderate and large risk factor space of $d = 33$ blood cell traits and $d = 92$ NMR metabolites we employ the stochastic search with 10,000 iterations and $p = 0.1$. This reflects an expected a priori model size of six for the small risk factor space and around three for the blood cell traits and nine for the high-dimensional NMR metabolite setting. The prior variance $\sigma^2$ is fixed to 0.25.

**Data pre-processing for NMR metabolites for simulation.** The first data resource used for the simulation and application is publicly available summarized data on genetic associations with risk factors derived from a NMR metabolite GWAS[11] from http://computationalmedicine.fi/data#NMR_GWAS. All of the metabolites were inverse rank-based normal transformed, so the association estimates are all in standard deviation units. In order to avoid selection bias, we choose genetic variants based on an external dataset. As the majority of the metabolite measures relates to lipids, we take $n = 150$ independent genetic variants that are associated with any of three composite lipid measurements (LDL cholesterol, triglycerides, or HDL cholesterol) at a genome-wide level of significance ($p$-value (two-sided) $<5 \times 10^{-8}$) in a large meta-analysis of the Global Lipids Genetics Consortium[19]. We extract beta-coefficients and standard errors of genetic associations for the 150 genetic variants and the 118 available metabolites. Next, we compute the genetic correlation structure between metabolites based on the $n = 150$ instrumental variables and exclude at random one of each pair of metabolites that are in stronger correlation than $|r| > 0.99$. For the simulation study each risk factor is scaled to have unit variance so all risk factors have an equal prior chance of being selected. Our final dataset $\beta_X$ for the simulation study comprises associations of $d = 92$ metabolites measured on $n = 150$ genetic variants. This allows us to investigate risk factor selection for a realistic genetic correlation structure between metabolites (Supplementary Fig. 2) and distribution of the regression coefficients.

**Data pre-processing for blood cell traits for simulation.** As a secondary data resource, we use publicly available summary data from the GWAS catalog https://www.ebi.ac.uk/gwas/ on 36 blood cell traits measured on nearly 175,000 participants[4]. Using all genetic variants that were genome-wide significant for any blood cell trait we have $n = 2667$ genetic variants as instrumental variables. There were eight pairs of blood cell traits with genetic correlation >0.99. After removing three composite traits (sum of eutrophil and eosinophil counts, granulocyte count, and sum of basophil and neutrophil counts) from further analysis, there was no pair of blood cell traits with greater genetic correlation than 0.99. The respective correlation matrix is shown in Supplementary Fig. 11. The final dataset used for the simulation consists of $d = 33$ blood cell traits as potential risk factors measured on $n = 2667$ genetic variants (pruned at $r^2 < 0.8$). For the simulation study each risk factor is scaled to have unit variance so all risk factors have an equal prior chance of being selected. We consider all $d = 33$ risk factors jointly for the simulation and consequently the simulation study has a realistic correlation structure between genetic associations of various blood cell traits (Supplementary Fig. 11) and a realistic distribution of regression coefficients.

**Data pre-processing and analysis for applied example of age-related macular degeneration.** In the applied example we demonstrate how MR-BMA can be used to select metabolites as causal risk factors for age-related macular degeneration (AMD). As risk factors we consider a range of circulating metabolites measured by NMR spectroscopy[11]. We use the same lipid-related genetic variants as in the simulation study. We restrict the risk factor space to include only lipoprotein measurements on total cholesterol content, triglyceride content, and particle diameter. For the various fatty acid measurements, we only included total fatty acids. Other lipid characteristics were highly correlated with the selected lipid measurements and including all of the lipid measurements would introduce multi-collinearity (RF2). As a next step we excluded all metabolite measures that did not have a single genetic variant that is genome-wide significant to meet the relevance criterion RF1. None of the remaining $d = 30$ metabolite measures have correlations in their genetic associations of $|r| > 0.985$ (Supplementary Fig. 2). Genetic associations with the outcome are taken from the latest GWAS meta-analysis on AMD[9] including $16,144$ patients and $17,832$ controls and is available from http://csg.sph.umich.edu/abecasis/public/amd2015/. To synchronise the genetic data on the metabolite risk factors and the AMD outcome, we match the effect alleles and we remove two genetic variants missing in the AMD data, so that the overall analysis includes $n = 148$ variants. Finally, we use the Ensembl Variant Effect Predictor[44] to annotate the genetic variants to the gene that is most likely affected.

We run MR-BMA including all $n = 148$ available genetic variants on the $d = 30$ metabolite associations using $p = 0.1$ as prior probability, $\sigma^2 = 0.25$ as prior variance, a maximum model size of 12 risk factors, and with 100,000 iterations in the shotgun stochastic search. To check the impact of the prior choice we first vary the prior probability (Supplementary Table 10) of selecting a risk factor from $p = 0.01$ to 0.3 reflecting 0.3–9.0 expected causal risk factors. This choice alters the posterior probabilities of various individual models, but the overall marginal inclusion probabilities of the risk factors are relatively stable. Finally, we vary the prior variance $\sigma^2$ from 0.01 to 0.49, which does not change the ranking (Supplementary Table 11).

**Reporting summary.** Further information on research design is available in the Nature Research Reporting Summary linked to this article.

## Data availability
All data used in our study is in the public domain. Our study is based on publicly available summary-level data on genetic associations from the International AMD Genetics consortium http://amdgenetics.org/, the GWAS catalog https://www.ebi.ac.uk/gwas/, and MAGNETIC NMR-GWAS http://www.computationalmedicine.fi/data. Pre-processed summary-level data used as input for this study is available from https://github.com/verena-zuber/demo_AMD.

## Code availability
R-code for MR-BMA (R version > 3.4.2, MIT license) and all other multivariable MR approaches (IVW, lars, lasso and elastic net) is provided on https://github.com/verena-zuber/demo_AMD. Moreover, we provide markdown scripts and the summary-level data on AMD and NMR metabolites as presented in the applied example on https://github.com/verena-zuber/demo_AMD. This allows the reader to reproduce all results and figures of the applied example and adapt the presented methodology on risk factor selection in multivariable MR into their own research.

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

## Acknowledgements

This work was supported by the UK Medical Research Council (MC_UU_00002/7). S.B. and V.Z. are supported by Sir Henry Dale Fellowship jointly funded by the Wellcome Trust and the Royal Society (Grant Number 204623/Z/16/Z). This study would not have been possible without the access to publicly available summary data. We would like to thank the International AMD Genetics consortium (http://amdgenetics.org/), the authors of the blood trait GWAS as curated by the GWAS catalog (https://www.ebi.ac.uk/gwas/), and the authors of the NMR-GWAS (http://www.computationalmedicine.fi/data).

## Author contributions

V.Z. and S.B. conceived and designed the study. V.Z carried out the statistical and computational analyses. J.M.C. and C.K. contributed to the design of the study and the interpretation of the findings. The paper was written by V.Z. and S.B.; and revised by all the co-authors. All co-authors have approved of the final version of the paper.

## Competing interests

The authors declare no competing interests.
