## [Peer Review File · Nature Communications]

Editorial Note: This manuscript has been previously reviewed at another journal that is not operating a transparent peer review scheme. This document only contains reviewer comments and rebuttal letters for versions considered at *Nature Communications*. Mentions of the other journal have been redacted.

Reviewers' Comments:

Reviewer #1:

Remarks to the Author:

This paper proposes a method for detecting causal risk factors using a Bayesian model averaging approach when there are multiple risk factors potentially associated with the outcome but only a few true causal risk factors.

It appears to me that the assumption that the true causal risk factors are sparse is potentially very important to this method. To what extent is this true? Have the authors done any simulations where this assumption is relaxed and more of the risk factors are true causal risk factors? I would like to see more discussion of this assumption and its potential implications throughout the paper. In the discussion you state that MR-BMA does not give unbiased estimates. This should also be stated in the results section when the MSE is being discussed. It would be helpful for the reader to have more discussion on the difference between the different results and results for the bias from the simulations as well as the MSE so that the difference between the estimation methods is clearer to the reader and a full comparison between the methods can be made.

What happens to the bias under the null? Are the results from MR-BMA biased if there is no true causal effect? This is not discussed and is very important to the strength of the method to detect causal risk factors.

More minor comments:

Although the discussion is clear that this is a method for risk factor selection rather than effect estimation this was not clear to me earlier in the paper – as this is an important factor for the paper it should be explicitly stated throughout.

Bottom of page 2: Please explain what you mean by Multivariable MR 'accounts for measured pleiotropy' and the conditions under which this is true.

Page 19 – weak instrument bias only tends to zero sample size increases (everything else held constant) as increasing the sample size increases the instrument strength. Consider rephrasing this to make it clear that increasing the sample size only decreases bias through increasing the instrument strength.

In the discussion you also state that selection is only possible if there are genetic variants that are predictors of these risk factors – does your method remove weakly predicted exposures from the estimation? If so this should be clearly explained and if not more explanation needs to be given about how weakly predicted exposures are removed from the model rather than give biased estimates of the effect.

Reviewer #3:

Remarks to the Author:

The authors have done a good job of addressing the reviewer comments, my own and those of others. They have made a strong effort to further clarify strengths and weaknesses of the method, and have added a new example using high dimensional RBC trait data. The added text comparing this approach to other MR methods is also appreciated.

Reviewer #4:

Remarks to the Author:

Zuber et al. proposed a new method for multivariable MR for high-throughput experiments based on Bayesian model averaging for variable selection.

I think the authors have replied thoroughly to previous reviewers' comments. Overall, I think this manuscript is mature and addressed the advantages with the limitations of the MR-BMA method. I only have a couple of minor comments/suggestions.

1. I think this is really important to convey to the readers the most critical limitation of MR-BMA, that this is a method for variable selection and will give biased effect estimates. I think this point is addressed in the first paragraph in discussion. However, I would like to see a stronger emphasis on the biased effect size estimates from MR-BMA (and all other variable selection methods). I think this will reduce the chance of MR-BMA being applied to effect size estimation, which is something that (I believe) the authors definitely want to avoid. I think it might also be worth mentioning in the discussion that 1) refit models to get effect estimates after variable selection will also give biased estimates (like the authors' reply to a previous comment) and 2) "selective inference" – inference for effect sizes after variable selection – is an active research area and is very relevant to MR-BMA and could be a future direction for variable selection based multivariable MR methods. (ref. Jonathan Taylor and Robert J. Tibshirani 2015 PNAS.

<https://www.pnas.org/content/112/25/7629.short>)

2. I am not entirely convinced that RF1 is the ultimate answer to avoid biased inference in MR-BMA (or any variable selection method in multivariable MR context) due to weak instruments. RF1 sets a sort of lower bound that is intuitively correct and understandable. Of course, a risk factor without any valid instrument should be excluded from the multivariable MR analysis. However, as shown and discussed by the authors, the number and strength of valid instruments for each risk factor will affect the power for a risk factor to be identified in variable selection. I am worried that the readers will take RF1 as THE criteria for instrument and risk factor selection and would suggest the authors to add some discussion/reminder to the readers in the method section (Choosing genetic variants as instruments) the importance of instrument selection and study design and how they can affect the results.

3. The authors mentioned briefly that they implemented a shotgun stochastic search algorithm to avoid exhaustive search for the entire model space for MR-BMA, which resulted in a much-improved computation cost and made the MR-BMA computationally feasible. However, the shotgun stochastic search algorithm relies on the sparsity of the data to reduce computation cost. I'm wondering if the computation cost will be prohibitive even with this algorithm when there are considerable proportions of risk factors that have true effects on the outcome. For example, if there are 41 metabolites (instead of 8) that have effects on the outcome, would the algorithm still complete the search within a reasonable time frame? Also, there's no mention of a potential limit for the number of risk factors that MR-BMA can handle. I think such a limit exists and might be investigated in previous literature for the BMA method in general. I think this information will benefit the reader when they try to apply this method for real data analysis.

Reviewers' comments:

Reviewer #1 (Remarks to the Author):

This paper proposes a method for detecting causal risk factors using a Bayesian model averaging approach when there are multiple risk factors potentially associated with the outcome but only a few true causal risk factors.

Reply: Thank you for your time reviewing our revised manuscript and the constructive feedback. Please find below our point by point replies to your comments.

1. It appears to me that the assumption that the true causal risk factors are sparse is potentially very important to this method. To what extent is this true? Have the authors done any simulations where this assumption is relaxed and more of the risk factors are true causal risk factors? I would like to see more discussion of this assumption and its potential implications throughout the paper.

Reply: We apologize for the imprecise language relating to the word “sparsity”. Our method assumes that there are several risk factors that have shared genetic predictors (otherwise it is better to perform separate analyses for individual risk factors), and that some of the risk factors are causal for the outcome and others are not. If all the risk factors are causal for the outcome, then existing methods (namely, the multivariable inverse-variance weighted method) are perfectly adequate and provide unbiased estimates.

The main motivation for our method is to identify which risk factors are causal for the outcome and to select the most likely causal risk factors from a set of candidate risk factors. Our method assumes that there are some true causal risk factors (signal) and some risk factors which do not have an effect (noise). In low-dimensional settings there is no necessity to assume sparsity or to specify a certain ratio of signal to noise variables for MR-BMA. We already conducted a simulation study in a scenario where 8 of the 12 risk factors are truly causal (Setting B) – this is already an example where the model is non-sparse (there are more signal variables than noise variables). MR-BMA performs well in this scenario and is not outperformed by other methods.

There are certain restrictions that arise when working on high-dimensional data settings with many risk factors (>20 risk factors). An exhaustive evaluation of all possible combinations of risk factors becomes computationally intractable for more than around 20 risk factors. In order to scale MR-BMA up to a larger set of risk factors we have implemented a shotgun stochastic search. The shotgun stochastic search evaluates combinations of risk factors with some evidence and avoids visiting combinations of risk factors with little evidence. For the design of the algorithm it was necessary to specify an upper limit for the model size, which is set to $k_{max}=12$ as optional parameter in the algorithm, but may be increased if the initial search shows that there is indeed evidence for models including 12 risk factors. In the application example considered in the paper, there was no strong evidence for models with more than three risk factors, so this was not a limitation in this particular application.

We thank the reviewer for bringing up this very important point and apologise again for being imprecise. We adjusted the manuscript accordingly and corrected references to sparsity and only mention the concept of sparsity when referring to high-dimensional data settings.

We have added the following text to the section “Risk factor selection as variable selection in the linear model”:

“Our implicit prior belief is that not all of the risk factors are causally related to the outcome and that there are some true causal risk factors (signal) and some risk factors which do not have an effect (noise).”

We have added the following paragraph to the discussion:

“MR-BMA is the first multivariable MR approach that can analyse a high-dimensional set of risk factors. When analysing many risk factors jointly one important implicit assumption of MR-BMA is sparsity i.e. the proportion of true causal risk factors compared to all risk factors considered is small. Since MR-BMA evaluates all possible combinations of risk factors exhaustively or all relevant combinations of risk factors in a shotgun stochastic search there is an upper bound for the maximum model size in order to keep the computation tractable. Sparsity is a common assumption for high-throughput data and we have seen in the applied example that the best models only contained one to three metabolites as risk factors despite allowing for a model size of up to twelve risk factors. Yet this is an important aspect of the algorithm and the maximum model size should be adjusted if models including many risk factors are expected or evidenced in the data.”

2. In the discussion you state that MR-BMA does not give unbiased estimates. This should also be stated in the results section when the MSE is being discussed. It would be helpful for the reader to have more discussion on the difference between the different results and results for the bias from the simulations as well as the MSE so that the difference between the estimation methods is clearer to the reader and a full comparison between the methods can be made.

Reply: We agree with the reviewer that it would be helpful to additionally report the bias in the simulation study. In accordance with the following comment 3, we have computed the bias separately for risk factors with positive causal effect, risk factors with negative causal effect (Simulation setting B) and no causal effect, in order to interpret the direction and magnitude of the bias.

We find that IVW is indeed the only unbiased causal effect estimate when there is a causal effect. All other methods which are designed for high-dimensional data settings are biased towards the Null and provide a conservative estimate which underestimates the magnitude of the true causal effect. In contrast, when there is no causal effect all methods are nearly unbiased with some of the novel methods for high-dimensional data even having a lower bias (closer to the Null) than the IVW estimate.

Accordingly we have added additional material on the bias (Supplementary Table 1) and the mean and standard deviation of the causal effect estimates (Supplementary Table 2 (Setting A) and 3 (Setting B)) and the following text to the Results:

“As can be seen from Supplementary Table 1, all estimation methods except the IVW are biased conservatively towards the Null when there is a true causal effect. Yet, all causal effect estimates are unbiased when there is no causal effect. Supplementary Table 2 (Setting A) and Supplementary Table 3 (Setting B) provide the mean and the standard deviation of the causal effect estimates, which confirm the large standard deviation of the IVW estimate compared to the other approaches.”

“Similarly to earlier results on the bias of the effect estimates we find that the IVW is unbiased when there is a causal effect, while the other methods designed for high-dimensional settings are conservatively biased towards the null, and only unbiased when there is no causal effect (Supplementary Table 1).”

3. What happens to the bias under the null? Are the results from MR-BMA biased if there is no true causal effect? This is not discussed and is very important to the strength of the method to detect causal risk factors.

Reply: We thank the reviewer for bringing up this extremely important distinction between bias under the null and bias under a causal effect. The important aspect of MR-BMA (and other high-dimensional approaches) is that the bias is towards the Null irrespective of if there is a causal effect or not. Consequently, MR-BMA is unbiased under the Null (No causal effect) and underestimates the magnitude of the causal effect otherwise (Causal effect). Overall this results in a more conservative estimation of the causal effect estimates and a reduced variance.

As illustrated already in the boxplots in Supplementary Figures 5&6 (Scenario A: 12 metabolites), 9&10 (Scenario B: 92 metabolites) and 14&15 (Scenario C: 33 blood cell traits), the simulations show that all methods are unbiased under the Null, but underestimate the magnitude of the effect when there is a true causal effect. For more clarity we have added Tables 1 (bias) and Table 2 and 3 (mean and standard deviation) to the Supplementary material.

We have added the following text to the Discussion section:

“This results in causal effect estimates being biased towards the Null when there is a causal effect and unbiased estimates when there is no causal effect.”

More minor comments:

a) Although the discussion is clear that this is a method for risk factor selection rather than effect estimation this was not clear to me earlier in the paper – as this is an important factor for the paper it should be explicitly stated throughout.

Reply: Indeed, we do not want to encourage the use of MR-BMA to estimate effect sizes. Model average causal effect estimates are provided, but they should only be used for comparison between different risk factors and to establish the direction of effect. We have added the following text to the Section “Multivariable Mendelian randomization and risk factor selection”

“MR-BMA reports the model averaged causal effects (MACE), representing conservative estimates of the direct causal effect of a risk factor on the outcome averaged across these models. These estimates can be used to compare risk factors or to interpret effect directions, but should not be interpreted absolutely. As we show in a simulation study based on real biomarker data, MR-BMA provides effect estimates biased towards the Null when there is a causal effect but reduces the variance of the estimate, trading bias for reduced variance. Consequently, MR-BMA enables a better and more stable detection of the true causal risk factors than either the conventional IVW method or other variable selection methods.”

And to the section “Causal estimation”:

“Both $\hat{\theta}_{\gamma}$ and $\hat{\theta}_{MACE}$ are conservative estimates of the true causal effect. They are biased towards the Null if there is a causal effect and unbiased otherwise. We therefore only recommend to interpret the direction of effect and the magnitude of these causal effect estimates in comparison with the effects of other risk factors.”

b) Bottom of page 2: Please explain what you mean by Multivariable MR ‘accounts for measured pleiotropic’ and the conditions under which this is true.

Reply: Standard MR requires genetic variants to be specific in their associations with a single risk factor of interest, and does not allow genetic variants to have pleiotropic effects on other risk factors on competing causal pathways. Multivariable MR allows genetic variants to be associated with multiple risk factors, provided these risk factors are measured and included in the analysis. Hence multivariable MR allows for ‘measured pleiotropic effects (Burgess and Thompson, 2015; Burgess, Dudbridge and Thompson, 2015; Sanderson *et al.*, 2018).

We have added this as an additional text to the Section “Multivariable Mendelian randomization and risk factor selection”.

c) Page 19 – weak instrument bias only tends to zero sample size increases (everything else held constant) as increasing the sample size increases the instrument strength. Consider rephrasing this to make it clear that increasing the sample size only decreases bias through increasing the instrument strength.

Reply: We thank the reviewer - that is an excellent suggestion. We have rephrased this paragraph accordingly to:

“However, in multivariable MR, weak instrument bias can be in any direction (Methods), although bias will tend to zero as the sample size increases and consequently the instrument strength increases.”

d) In the discussion you also state that selection is only possible if there are genetic variants that are predictors of these risk factors – does your method remove weakly predicted exposures from the estimation? If so this should be clearly explained and if not more explanation needs to be given about how weakly predicted exposures are removed from the model rather than give biased estimates of the effect.

Reply: The general MR framework states that a genetic variant is only a valid instrumental variable if it can predict the exposure (IV1 Relevance assumption). In turn, multivariable MR can only include risk factors that are predicted by the genetic variant included as instrumental variables. It is the analyst’s responsibility to ensure that adequate genetic predictors are available for all risk factors that are included in the analysis. We make this assumption explicit by formulating the following assumption on the risk factors that may be included, in particular:

- RF1 Relevance: The risk factor needs to be strongly instrumented by at least one genetic variant included as instrumental variable.

References:

Burgess, S., Dudbridge, F. and Thompson, S. G. (2015) ‘Re: “Multivariable Mendelian randomization: the use of pleiotropic genetic variants to estimate causal effects”’, *American Journal of Epidemiology*. Oxford Univ Press, 181(4), pp. 290–291.

Burgess, S. and Thompson, S. G. (2015) ‘Multivariable Mendelian randomization: the use of pleiotropic genetic variants to estimate causal effects’, *Am J Epidemiol*, 181(4), pp. 251–260. doi: 10.1093/aje/kwu283.

Sanderson, E. *et al.* (2018) ‘An examination of multivariable Mendelian randomization in the single-sample and two-sample summary data settings’, *International Journal of Epidemiology*. doi: 10.1093/ije/dyy262.

Reviewer #3 (Remarks to the Author):

The authors have done a good job of addressing the reviewer comments, my own and those of others. They have made a strong effort to further clarify strengths and weaknesses of the method, and have added a new example using high dimensional RBC trait data. The added text comparing this approach to other MR methods is also appreciated.

Reply: We thank the reviewer for his/her comments and help to improve our manuscript.

Reviewer #4 (Remarks to the Author):

Zuber et al. proposed a new method for multivariable MR for high-throughput experiments based on Bayesian model averaging for variable selection.

I think the authors have replied thoroughly to previous reviewers' comments. Overall, I think this manuscript is mature and addressed the advantages with the limitations of the MR-BMA method. I only have a couple of minor comments/suggestions.

Reply: Thank you for your time reviewing our manuscript and the constructive feedback. Please find below our point by point replies to your comments.

1. I think this is really important to convey to the readers the most critical limitation of MR-BMA, that this is a method for variable selection and will give biased effect estimates. I think this point is addressed in the first paragraph in discussion. However, I would like to see a stronger emphasize on the biased effect size estimates from MR-BMA (and all other variable selection methods). I think this will reduce the chance of MR-BMA been applied to effect size estimation, which is something that (I believe) the authors definitely want to avoid. I think it might also worth mentioning in the discussion that 1) refit models to get effect estimates after variable selection will also give biased estimates (like the authors' reply to a previous comment) and 2) "selective inference" – inference for effect sizes after variable selection – is an active research area and is very relevant to MR-BMA and could be a future direction for variable selection based multivariable MR methods. (ref. Jonathan Taylor and Robert J. Tibshirani 2015 PNAS.

<https://www.pnas.org/content/112/25/7629.short>

Reply: We agree with the reviewer that this is a very interesting future direction that needs to be explored in the context of multivariable MR. We have added accordingly the following text to the Discussion:

"One necessary future development is post-selection inference (Taylor and Tibshirani, 2015; Lee et al., 2016) in the high-dimensional multivariable MR framework. MR-BMA does not provide unbiased causal effect estimates. Re-fitting an unbiased multivariable MR model after risk factor selection would ignore the uncertainty of the selection and consequently not provide valid inferences."

2. I am not entirely convinced that RF1 is the ultimate answer to avoid biased inference in MR-BMA (or any variable selection method in multivariable MR context) due to weak instruments. RF1 sets a sort of lower bound that is intuitively correct and understandable. Of course, a risk factor without any valid instrument should be excluded from the multivariable MR analysis. However, as shown and discussed by the authors, the number and strength of valid instruments for each risk factor will affect the power for a risk factor to be identified in variable selection. I am worried that the readers will take RF1

as THE criteria for instrument and risk factor selection and would suggest the authors to add some discussion/reminder to the readers in the method section (Choosing genetic variants as instruments) the importance of instrument selection and study design and how they can affect the results.

Reply: We thank the reviewer for bringing up this point. We already mention in the manuscript that the selection of risk factors may depend on which genetic variants were included as instrumental variables. It is indeed possible that some risk factors may be downweighted if there are no strong instruments to predict them. For the interpretation it is important to use MR-BMA to prioritise risk factors, and not to interpret low posterior probabilities as strong evidence for no causal effect as we mention in the discussion (“Caution is needed for the interpretation of null findings, particularly in our example for non-lipid risk factors, as these might be deprioritised in terms of statistical power by our choice of genetic variants.”).

We have added the following text to the discussion in order to highlight this highly important aspect even more:

“The instrument selection and general study design are essential for the MR-BMA approach and we strongly recommend the user to be critical in the choice of genetic variants and risk factors. Moreover, similar to standard MR we urge to perform model checks and be transparent in the presentation of the removal of outlier/influential genetic variants.”

And the following paragraph to the Section “Choosing genetic variants as instruments”:

“The study design and in particular the selection of genetic variants as IVs are the most important steps for multivariable MR and great care needs to be taken when designing the study and also when reporting the study design. All interpretation of the results is conditional on the genetic variants selected as IVs.”

3. The authors mentioned briefly that they implemented a shotgun stochastic search algorithm to avoid exhaustive search for the entire model space for MR-BMA, which resulted in a much-improved computation cost and made the MR-BMA computationally feasible. However, the shotgun stochastic search algorithm relies on the sparsity of the data to reduce computation cost. I’m wondering if the computation cost will be prohibitive even with this algorithm when there are considerable proportion of risk factors have true effects on the outcome. For example, if there are 41 metabolites (instead of 8) have effects on the outcome, would the algorithm still complete the search within reasonable time frame? Also, there’s no mention of a potential limit for the number of risk factors that MR-BMA can handle. I think such limit exists and might be investigated in previous literature for BMA method in general. I think this information will benefit the reader when they try to apply this method for real data analysis.

Reply: Interestingly, a quick literature research did not provide any precise recommendation on the maximum model size in shotgun stochastic search. For

example the original publication shows that for the applied example only models up to a size of seven variables are explored (Hans, Dobra and West, 2007).

A more recent methodological article presenting a simplified shotgun stochastic search demonstrated an average model size of 20 when analysing 2000 variables from a gene expression experiment (Shin, Bhattacharya and Johnson, 2018).

For example finemap (Benner *et al.*, 2016), a tool to finemap genomic regions in the search for causal genetic variants limits the maximum model size to five when exploring regions including hundreds or thousands of genetic variants.

We performed an analysis of the runtime of our algorithm (using niter=1000 repetitions for the stochastic search) on the NMR metabolite data (including $d=92$ risk factors as variables) depending on the number of true causal risk factors simulated (#causal variables ranging from 2 to 20) and the maximum model size allowed (kmax, ranging from 2 to 20). The table below shows the median of the runtime (in seconds) over three repetitions. The runtime increases when the number of causal variants and/or the allowed maximum model size increases.

#causal variables	kmax=									
	2	4	6	8	10	12	14	16	18	20
2	21.14	43.73	56.42	99.70	134.01	192.02	283.20	219.78	308.42	372.66
4	18.11	44.18	65.52	116.73	169.53	217.03	206.92	253.24	303.80	432.35
6	17.67	52.75	86.11	117.09	141.55	258.61	324.90	377.14	329.87	313.06
8	18.95	57.91	113.41	164.26	187.06	183.76	259.38	265.64	358.95	463.06
10	17.55	57.14	108.37	138.86	178.69	212.08	276.72	346.70	287.04	419.45
12	17.63	56.72	108.71	137.53	152.69	265.46	308.42	326.45	343.76	361.12
14	17.86	67.77	85.89	131.87	204.54	229.46	226.78	347.65	347.57	473.85
16	18.33	69.26	87.98	147.15	201.48	217.68	253.32	347.27	390.37	362.38
18	15.88	62.00	105.41	131.28	157.89	292.89	249.94	348.55	342.63	390.00
20	15.25	70.05	121.80	131.48	186.53	224.38	318.72	277.28	324.29	522.17

We agree with the reviewer that this is an important aspect that needs to be mentioned and we have added the following paragraph to the discussion:

“Since MR-BMA evaluates all possible combinations of risk factors exhaustively or all relevant combinations of risk factors in a shotgun stochastic search there is an upper bound for the maximum model size in order to keep the computation tractable. Sparsity is a common assumption for high-throughput data and we have seen in the applied example that the best models only contained one to three metabolites as risk factors despite allowing for a model size of up to twelve risk factors. Yet this is an important aspect of the algorithm and the maximum model size should be adjusted if models including many risk factors are expected or evidenced in the data.”

References:

Benner, C. *et al.* (2016) ‘FINEMAP: Efficient variable selection using summary data from genome-wide association studies’, *Bioinformatics*. doi:

10.1093/bioinformatics/btw018.

Hans, C., Dobra, A. and West, M. (2007) 'Shotgun Stochastic search for "Large p" regression', *Journal of the American Statistical Association*, 102(478), pp. 507–516. doi: 10.1198/016214507000000121.

Lee, J. D. *et al.* (2016) 'Exact post-selection inference, with application to the lasso', *Annals of Statistics*. doi: 10.1214/15-AOS1371.

Shin, M., Bhattacharya, A. and Johnson, V. E. (2018) 'Scalable Bayesian variable selection using nonlocal prior densities in ultrahigh-dimensional settings', *Statistica Sinica*. doi: 10.5705/ss.202016.0167.

Taylor, J. and Tibshirani, R. J. (2015) 'Statistical learning and selective inference', *Proceedings of the National Academy of Sciences of the United States of America*. doi: 10.1073/pnas.1507583112.

Reviewers' Comments:

Reviewer #1:

Remarks to the Author:

I am happy with the changes the authors have made to the manuscript.

Reviewer #4:

Remarks to the Author:

The authors addressed my previous comments with more detailed discussions and additional analyses (not included in the paper). I think the authors presented the method clearly with enough attention to limitations and the intended use of their method. I have no further comments.

REVIEWERS' COMMENTS:

Reviewer #1 (Remarks to the Author):

I am happy with the changes the authors have made to the manuscript.

Reply: We thank the reviewer for his/her time and effort to improve the manuscript.

Reviewer #4 (Remarks to the Author):

The authors addressed my previous comments with more detailed discussions and additional analyses (not included in the paper). I think the authors presented the method clearly with enough attention to limitations and the intended use of their method. I have no further comments.

Reply: We thank the reviewer for his/her positive comments and help to improve the manuscript.